# Fruit Pouch Consumption and Dietary Patterns Related to BMIz at 18 Months of Age

**DOI:** 10.3390/nu13072265

**Published:** 2021-06-30

**Authors:** Ellen Lundkvist, Elisabeth Stoltz Sjöström, Richard Lundberg, Sven-Arne Silfverdal, Christina E. West, Magnus Domellöf

**Affiliations:** 1Department of Clinical Sciences, Pediatrics, Umeå University, 901 85 Umeå, Sweden; ellen.lundkvist@gmail.com (E.L.); richard.lundberg@umu.se (R.L.); sven.arne.silfverdal@umu.se (S.-A.S.); christina.west@umu.se (C.E.W.); 2Department of Food, Nutrition and Culinary Science, Umeå University, 901 87 Umeå, Sweden; elisabeth.stoltz.sjostrom@umu.se

**Keywords:** childhood obesity, baby food pouches, fruit pouches, fruit juice, sugar-sweetened beverages, whole fruit, breastfeeding, milk cereal drink, maternal BMI, gestational weight gain

## Abstract

Concerns have been raised that an overconsumption of baby food fruit pouches among toddlers might increase the risk of childhood obesity. This study aimed to quantify the consumption of fruit pouches and other fruit containing food products and to explore potential correlations between the consumption of these products and body-mass index *z*-score (BMIz) at 18 months, taking other predictive factors into consideration. The study was based on 1499 children and one-month-recall food frequency questionnaires from the Swedish population-based birth cohort NorthPop. Anthropometric outcome data were retrieved from child health care records. BMIz at 18 months of age was correlated to maternal BMI and gestational weight gain and inversely correlated to fruit juice consumption and breastfeeding. BMIz at 18 months of age was not correlated to consumption of fruit pouches, sugar-sweetened beverages, whole fruit or milk cereal drink. Overweight at 18 months of age was correlated to maternal BMI and inversely correlated to breastfeeding duration. To our knowledge, this is the first study that investigates possible associations between baby food fruit pouch consumption and overweight in toddlers. We found that moderate fruit pouch consumption is not associated with excess weight at 18 months of age.

## 1. Introduction

The global burden of childhood obesity has grown substantially since the 1990s, but has plateaued in many high-income countries [1,2]. In 2018, the prevalence of overweight and obesity in Swedish six- to nine-year-old children had reached 21% [3]. Obese children are more likely than normal weight children to grow up to become obese adults [4]. Furthermore, childhood obesity is associated with anxiety and depression in childhood [5] and increased risk of coronary heart disease, diabetes type 2 and some forms of cancer in adulthood [6]. A healthy diet is fundamental for prevention of childhood obesity [7]. Thus, identifying modifiable risk factors for pediatric obesity as targets of interventional strategies could impact lifelong health.

Food pouches were introduced on the baby food market some 20 years ago [8]. The baby food pouches have gained popularity among children and parents and, in 2016, approximately one third of American children aged 6–12 months consumed at least one pouch per week [9]. Most of these baby food pouches contain fruit purées (fruit pouches) [10]. The European Society for Paediatric Gastroenterology, Hepatology and Nutrition (ESPGHAN) recommends that the energy intake from free sugars for children should be less than 5% of total energy intake [11]. Contrary to these recommendations, baby food pouches are more likely to contain higher levels of sugar than other package types [10,12,13,14]. Particular concern has been raised that fruit pouches are too energy dense, contain too much sugar and induce a preference for sweet taste. In turn, these features are suggested to result in overfeeding and excessive weight gain in toddlers [14,15]. Yet, health outcomes in relation to consumption of fruit pouches have been insufficiently studied [16].

Childhood obesity is attributed to genetic susceptibility mediated by a number of prenatal and early childhood dietary variables [17]. A recognized early life dietary risk factor is the consumption of sugar-sweetened beverages (SSBs). In addition to recommendations on limiting the SSB consumption in early childhood, fruit juice is often discouraged as part of an unhealthy diet in toddlers [18]. Fruit juice is an established source of free sugars and considered a potential risk factor of overweight [11,18,19]. However, the relationship between fruit juice consumption and body-mass index (BMI) in children is inconsistent [20,21,22]. By contrast, whole fruit is thought of as healthy [23] and consumption is recommended due to its content of nutrients and fibers [18]. As fruit pouches are drunk, rather than eaten [16], they resemble fruit juices rather than servings of whole fruit. 

Other early life dietary variables linked to childhood overweight are breastfeeding and milk cereal drink (MCD) intake. Breastfeeding is associated with slower growth rate in infancy and is considered a protective factor [24,25,26,27,28]. In Sweden, MCDs are commonly consumed by infants and toddlers and studies have shown that MCDs are a risk factor for childhood overweight [29,30,31]. In addition, maternal pre-pregnancy BMI and gestational weight gain (GWG) are identified among the prenatal risk factors [17,27]. In addition, childhood obesity is inversely related to socioeconomic status (SES) in high-income countries [32] and low SES may be linked to rapid growth in infancy and later obesity [33]. 

### Aims of the Study 

The aims of this study were to quantify the consumption of baby food fruit pouches in a Swedish population-based cohort of children (the NorthPop birth cohort [34]) and to investigate possible associations between the intake of fruit pouches and the BMI-for-age *z*-score (BMIz) and overweight at 18 months of age. In addition, the associations between BMIz at 18 months and consumption of other fruit containing food products, previously recognized risk factors (MCD consumption, maternal BMI and GWG), the possible preventive effect of breastfeeding and potential SES variables were investigated. In addition, the potential correlation between BMIz at 18 months and SES was explored.

## 2. Materials and Methods 

This study was carried out within the research infrastructure of the population based, prospective NorthPop birth cohort study in northern Sweden [34]. Data were retrieved from the NorthPop questionnaires (NPQ) that included food frequency questionnaires (FFQ), the Swedish Pregnancy Register [35], the Longitudinal Integrated Database for Health Insurance and Labour Market Studies (LISA) provided by Statistics Sweden [36] and participants’ child health care records.

### 2.1. Study Population

Between May 2016 and August 2018, 4064 pregnant women within the NorthPop catchment area had their routine ultrasound examinations at Umeå University Hospital, usually in gestational week 17–18. At that visit they were invited to participate in the NorthPop study and enrolled. Inclusion criteria were pregnant woman aged 18 years or older, Swedish-speaking, viable pregnancy in gestational week 14–24 and intention to give birth and plan to reside within the catchment area in the following years. An additional inclusion criterion for this study was access to child health care record anthropometric data at 18 ± 1 months of age. The final study population consisted of 1499 children (Figure 1).

### 2.2. Exposure Variables: Fruit Pouch Consumption and Recognized Risk and Protective Factors

The NPQ provided nutrition and feeding data of the children at four, nine and 18 months of age. The data on consumption of fruit pouches, fruit juice, SSBs and whole fruit at 18 months of age and MCDs at nine and 18 months were collected through a one-month-recall FFQ. The FFQ questions can be found in Appendix B. The NPQ also supplied breastfeeding data at four, nine and 18 months of age.

Data on maternal BMI were retrieved from the Swedish Pregnancy Register. Maternal BMI was calculated when the pregnancy was registered at the first maternal health center appointment, usually at 7–12 weeks of gestation. To reflect pre-pregnancy BMI, only maternal BMI recorded until gestational week 16 was considered in this study. The GWG was self-reported in the NPQ sent to the mothers when their children were four months old.

The variables of SES included in this study were yearly household disposable income and the education levels of the parents. Data regarding these variables were obtained from the LISA database of Statistics Sweden. Household disposable income for the tax year the child was born (2016, 2017 or 2018) was used and accounted for in Swedish Krona (SEK). Statistics Sweden also provided household disposable income data of a reference population consisting of all households with children born 2016, 2017 and 2018 in the county of Västerbotten, Sweden. The education levels of the parents were adjusted to the Swedish education system and coded 1 to 7 in LISA [37]. The corresponding International Standard Classification of Education 2011 levels (ISCED) [38] are found in Appendix C.

### 2.3. Health Outcome Data: BMIz and Overweight 

The BMIz of the children were calculated from the weight, length, sex and age at the date measured (in number of days) using the WHO Anthro Survey Analyser tool [39]. The anthropometric measurements weight and length were retrieved from the child health care records at the 18 month follow up, described in the National Handbook for Child Health Services in Sweden [40]. Overweight was defined as the BMIz > 2 [41]. Overweight included those who were also obese, BMIz > 3.

### 2.4. Statistical Analyses

To assess possible differences between means of BMIz at 18 months of age for nominal variables, one-way analysis of variance (ANOVA) tests were used. To assess differences between means of BMIz at 18 months between dichotomous variables, independent *t*-tests were carried out. To determine possible correlations between continuous variables and BMIz at 18 months, Pearson’s correlation was used. To assess correlations between BMIz at 18 months and ordinal variables, Spearman’s correlation was used. The variables that were significantly associated with BMIz at 18 months of age were tested in a multivariate regression model using the general linear model and a manual stepwise procedure.

To assess possible differences in means of continuous or ordinal variables between the groups of overweight and normal weight children at 18 months of age, independent *t*-tests were carried out. To assess significant associations between overweight at 18 months of age and nominal variables, Pearson’s chi-square tests were performed.

For a number of variables, the response options were merged to create categories more equal in size for the ANOVA and chi-square analyses. According to consumption frequency at 18 months of age, the children were categorized as nonconsumers, seldom consumers (less frequently than once per week) and regular consumers (once per week or more often) of fruit pouches, fruit juice and sugar-sweetened beverages. According to consumption frequency of MCDs at nine months of age, the children were categorized as nonconsumers, consumers (monthly and weekly consumption) and frequent consumers (daily consumption). The variable household disposable income the tax year the child was born was not normally distributed and transformed using the decadic logarithm. Variability was expressed as mean ± one standard deviation (SD). The level of significance was set to *p* < 0.05. Statistical analyses of data were carried out in SPSS Version 26.0 (IBM Corp. Armonk, NY, USA, 2019).

## 3. Results

### 3.1. Characteristics of the Study Population

The characteristics of the participating children and mothers are displayed in Table 1.

The socioeconomic characteristics considered were income and level of education. The mean yearly household disposable income the year the child was born was 526,058 SEK (*n* = 1491, missing data for *n* = 8). The means of yearly household disposable income for the reference groups were: 413,600 SEK (2016), 449,700 (2017) and 494,800 (2018). The average of these years was 452,700 SEK. Parental levels of education are displayed in Table 2.

An overview of NPQ response rates for all selected early life dietary variables are presented in Appendix A.

### 3.2. Fruit Pouches and Other Fruit Containing Foods and Drinks at 18 Months of Age

At 18 months of age, the majority of the children (76.5%) were nonconsumers or seldom consumers of fruit pouches. In total, 23.5% (*n* = 246) of the children had consumed fruit pouches once per week or more often i.e. regular consumers. The mean BMIz at 18 months of age was 0.59 ± 0.91 of nonconsumers (40.5%, *n* = 423), 0.72 ± 0.91 of seldom consumers (36.0%, *n* = 377) and 0.61 ± 0.97 of regular consumers. The distribution of fruit pouch consumption at 18 months is illustrated in Figure 2a and the means BMIz according to fruit pouch consumption levels in Figure 2b. Comparing fruit pouch consumption levels, a one-way ANOVA test did not show any significant difference between means BMIz at 18 months of the nonconsumers, seldom consumers or regular consumers.

The distribution of fruit juice consumption frequency at 18 months is presented in Figure 3a and the mean BMIz according to fruit juice consumption levels in Figure 3b. The mean BMIz were 0.72 ± 0.92 of nonconsumers, 0.61 ± 0.91 of seldom consumers and 0.45 ± 0.98 of regular consumers. Comparing the juice consumption levels, a one-way ANOVA revealed a significant difference between means BMIz at 18 months, F (2.10) = 5.82, *p* = 0.003. A Tukey post hoc test showed that the mean of BMIz at 18 months was significantly higher of nonconsumers than of regular consumers (*p* = 0.002). The mean BMIz at 18 months of the group seldom consumers did not differ significantly from the means of the other groups. A chi-square test detected no significant association between fruit juice consumption level and overweight at 18 months of age.

The distribution of SSB consumption frequency at 18 months is presented in Figure 4a and the mean BMIz according to fruit juice consumption levels in Figure 4b. The proportion of children who were regular consumers of SSBs (12.6%, *n* = 129) was notably lower than the proportion regular consumers of fruit pouches (23.5%). Comparing SSB consumption levels, a one-way ANOVA test did not show any significant difference between means BMIz at 18 months of nonconsumers, seldom consumers or regular consumers.

The distribution of whole fruit consumption at 18 months is displayed in Figure 5a. If categorized according to the same consumption levels as fruit pouches, the proportion of whole fruit regular consumers was 90.5% (*n* = 933). This was more than three times the proportion fruit pouch regular consumers (23.5%). These consumption levels were too uneven in size to test BMIz mean differences by carrying out an ANOVA. An independent *t*-test detected no significant difference between the mean BMIz at 18 months of age of daily whole fruit consumers and more seldom than daily consumers. The mean BMIz at 18 months of the latter was 0.62 ± 0.93 (*n* = 410, 39.8 %) compared to 0.66 ± 0.93 (*n* = 621, 60.2%) in daily whole fruit consumers (Figure 5b). The mean BMIz at 18 months of the nonconsumers was higher (0.94 ± 0.98) than of those who consumed whole fruit four to six times per day (0.57 ± 1.07), but the difference was not statistically significant.

### 3.3. Dietary Variables Before 18 Months of Age: Breastfeeding and Milk Cereal Drink Consumption 

Possible associations between dietary variables before 18 months of age and BMIz at 18 months were also explored. The variables considered were breastfeeding and MCD consumption. Breastfeeding, both at four and nine months of age, was associated with BMIz at 18 months of age. The distribution of children who were formula-fed or both formula and breastfed and exclusively breastfed is displayed in Figure 6a. The mean BMIz at 18 months of the children exclusively breastfed at four months, 0.62 ± 0.93, was significantly lower than the mean of the children who were formula fed or, both formula and breastfed, 0.76 ± 0.90, *t* (*n* = 1191) = 2.40, *p* = 0.016 (Figure 6b). A chi-square test detected no significant association between exclusive breastfeeding at four months and overweight at 18 months of age.

The distribution of children not breastfed and breastfed at nine months is displayed in Figure 7a. The mean BMIz at 18 months of those who were breastfed at nine months was significantly lower, 0.52 ± 0.90, than the mean of the those who were not breastfed at that age, 0.74 ± 0.92, *t* (*n* = 1175) = 4.21, *p* < 0.001 (Figure 7b). Moreover, the Pearson chi-square test showed that not breastfeeding at nine months of age was associated with overweight at 18 months of age, χ^2^ (1, *n* = 1177) = 6.25, *p* = 0.012).

The distribution of breastfeeding duration is presented in Figure 8. The mean breastfeeding duration was 8.8 ± 4.0 months. There was a significant inverse correlation between breastfeeding duration and BMIz at 18 months of age, ρ = −0.076, *p* = 0.012. Furthermore, the breastfeeding duration of the children who were overweight at 18 months (7.9 ± 3.4 months, *n* = 76) differed significantly from the children who were not overweight at 18 months (8.8 ± 4.0 months, *n* = 1034), *t* (1108) = 2.02, *p* = 0.044.

The distribution of MCD consumption at nine months is displayed in Figure 9a. A one-way ANOVA for MCD consumption levels at nine months of age detected a significant difference in means of BMIz at 18 months between nonconsumers, 0.57 ± 0.97 (*n* = 350, 30.9%) and frequent consumers 0.73 ± 0.90 (*n* = 593, 52.3%), F (2.11) = 4.42, *p* = 0.012. A Tukey post hoc test showed that the mean BMIz at 18 months of frequent consumers was higher than that of nonconsumers (*p* = 0.022). The mean BMIz at 18 months of consumers, 0.57 ± 0.87 (*n* = 190, 16.8%), did not significantly differ from any of the others (Figure 9b). The MCD consumption level at nine months was not significantly correlated to overweight at 18 months of age. 

The distribution of MCD consumption at 18 months is illustrated in Figure 10a. The consumption level of MCDs at 18 months was not associated with BMIz (Figure 10b).

### 3.4. Prenatal Variables: Maternal BMI and GWG

The maternal BMI significantly correlated with their children’s BMIz at 18 months of age (r = 0.122, *p* < 0.001). In addition, an independent variable *t*-test showed that the mean BMI of the mothers whose children were overweight at 18 months of age was significantly higher (26.07 ± 5.04, *n* = 90) than that of those whose children were not overweight at 18 months (maternal BMI 24.44 ± 4.12, *n* = 1243), *t* (106.9) = −3.02, *p* = 0.003.

The GWG correlated with the children’s BMIz at 18 months of age (ρ = 0.071, *p* = 0.014). The mean GWG of the mothers whose children were overweight at 18 months did not differ significantly from the mean GWG of the mothers whose children were not overweight at 18 months of age.

### 3.5. Socioeconomic Status: Yearly Household Disposable Income and Levels of Education

Notably, the mean yearly household disposable income of the study population was higher than that of the reference group. The decadic logarithm of the yearly household disposable income did neither correlate to BMIz at 18 months of age nor to fruit pouch consumption levels at 18 months of age. Furthermore, there was no significant difference between means of log yearly household disposable income between overweight and not overweight children at 18 months of age.

The education level of the mothers was at least secondary in 96.6% of cases. The majority of mothers had also completed three years of post-secondary education or more (Table 2). In this study, there was no correlation between maternal education level and the child’s BMIz at 18 months of age. Of the co-parents, 80.6% had completed three years of upper secondary education or higher. There was no correlation between co-parent education level and the child’s BMIz at 18 months of age.

Since yearly household disposable income and parental education levels were not associated with BMIz at 18 months of age, possible correlations to dietary variables were not included in the multivariate analyses.

### 3.6. Multivariate Analyses 

In the multivariate model, we chose to include only the breastfeeding variable and prenatal maternal anthropometric variable most significantly associated with BMIz at 18 months of age. These were breastfeeding at nine months of age and maternal BMI in early pregnancy. The first multivariate analysis was modelled to test interaction and included fruit juice consumption at 18 months, breastfeeding at nine months, MCD consumption at nine months and maternal BMI. In the multivariate analysis, the MCD consumption at nine months turned out not to be significantly correlated to BMIz at 18 months of age and, thus, excluded in the final model. The final analysis (Table 3) was modelled to test main effects and showed that the following variables were statistically significant independent predictors of BMIz at 18 months of age: Fruit juice consumption at 18 months, breastfeeding at nine months and maternal BMI, F (4) = 10.96, *p* < 0.001, *R*^2^ = 0.045 and adjusted *R*^2^ = 0.041.

## 4. Discussion

To our knowledge, this is first study that investigates possible associations between baby food fruit pouch intake and overweight in toddlers. In this study, fruit pouch, SSB, whole fruit and MCD consumption frequency at 18 months of age were not significantly associated with BMIz or overweight at 18 months of age. The variables that were significantly associated with BMIz at 18 months of age were: fruit juice consumption at 18 months (negative correlation), exclusive breastfeeding at four months (negative correlation), breastfeeding at nine months (negative correlation), breastfeeding duration (negative correlation), MCD consumption at nine months, maternal BMI and GWG. Of these, fruit juice consumption, breastfeeding and maternal BMI remained significant predictors in the multivariate model. Furthermore, maternal BMI, breastfeeding at nine months of age (negative correlation) and breastfeeding duration (negative correlation) were significantly correlated to overweight at 18 months of age.

This study showed that less than a fourth (23.5%) of the 18-month-olds in the NorthPop cohort consumed fruit pouches once per week or more often. This constitutes a lower proportion than the one third of the six-to-twelve-months-old children in a previous US study [9]. In the NorthPop study population, the overall fruit pouch consumption could be described as moderate as only 2.5% were served fruit pouches daily. In addition, the fruit juice and SSB consumption frequencies were moderate. By contrast, the daily consumption of whole fruit was 57% and the proportion of the study population that consumed whole fruit once per week or more often was 90.5%. These findings are in line with the Swedish Food Agency recommendations, discouraging sugar containing beverages and encouraging fresh fruit consumption [18]. Possibly, these findings could be interpreted as there is a general awareness of national recommendations among NorthPop cohort participants and an inclination to adhere to recommendations. In addition, the NorthPop study population may have been influenced by the criticism of baby food pouches in popular media [43]. At least, fruit pouches do not appear to have replaced whole fruit. Furthermore, we found no significant association between fruit pouch consumption level and BMIz or overweight at 18 months of age. These findings indicate that the majority of the children in the NorthPop cohort do not overconsume fruit pouches and that current consumption levels do not increase the risk of excessive weight gain. 

Whereas consumption of fruit pouches did not correlate to BMIz at 18 months of age, we found that regular consumption of fruit juice did. The significantly lower mean BMIz at 18 months of the children who consumed fruit juice at least once per week, compared with that of children who did not consume any fruit juice at all, may possibly be explained by a generally healthier diet. This could be viewed as consistent with previous findings that have linked fruit juice consumption to whole fruit consumption [21]. Yet, in this study, there were no association between whole fruit consumption and BMIz or overweight at 18 months of age among children in the NorthPop cohort. Higher consumption of SSBs and fruit juice in infancy has been shown to be associated with lower maternal socioeconomic status, including an education level lower than high-school degree [44]. Possibly, the modest consumption of SSBs and fruit juice in this study population reflected that the education level of the mothers was at least secondary in 96.6% of cases.

Consistent with much of existing evidence [24,25,27], this study showed that BMIz at 18 months of age was inversely associated with breastfeeding, both at four and nine months of age and the effect of breastfeeding at nine months remained in the multivariate model. Moreover, overweight at 18 months of age was associated with shorter duration of breastfeeding and not breastfeeding at nine months of age.

When tested in univariate analysis, MCD intake at nine months was associated with a higher BMIz at 18 months of age. However, this significant association did not remain in the multivariate analysis when breastfeeding was included in the model. We believe that breastfeeding is the actual explanatory variable since MCD consumption was significantly, negatively correlated with breastfeeding at nine months of age (not shown). Furthermore, MCD consumption at 18 months did not correlate to BMIz at that age. This is reassuring, as more than half of the study population was frequent consumers of MCDs both at nine and 18 months of age.

Both the prenatal risk factors maternal BMI and GWG correlated to BMIz at 18 months of age. Maternal BMI was the variable that contributed most impact according to the general linear model and was associated to overweight at 18 months of age.

Surprisingly, we found no association between parental education levels or yearly household disposable income and BMIz at 18 months of age. The former can probably be explained by the small proportion of low educational level in the NorthPop study population. The mothers’ level of education was primary in only 0.7% of cases and lower secondary in 2.7%. The corresponding figures for co-parents were only 0.4% and 3.0%. The fact that the study population mean of yearly household income was higher than the mean of all families in the region/county with children born 2016–2018, may reflect that the lowest income group was underrepresented in the study population. It is possible that the high incomes and high educational levels in this cohort have influenced the results, e.g., the unexpected observation that fruit juice consumption was inversely correlated with BMIz at 18 months.

Strengths of the study are the large study sample and the high response rates of the NPQ questions (approximately 70%). A limitation of FFQs in general is potential recall bias. Further, particular limitations of the NorthPop FFQ are thatit only included feeding frequency, not serving sizes and that the full FFQ is quite extensive and may be prone to respondent fatigue. Further, the NorthPop FFQ has not yet been validated. A validation of the full FFQ used in this ongoing cohort study is planned to be performed within the next years but is not yet available. However, as the focus of this study was not nutrient intakes but the intake frequencies of certain food products, the the FFQ was useful for identifying, e.g., nonconsumers and frequent consumers of certain food products.

Another limitation was that our study population included few families with low parental education level.

## 5. Conclusions

In the NorthPop cohort, the overall fruit pouch consumption at 18 months of age was moderate and not associated with excess weight gain. Further, regular fruit juice consumption on a weekly basis was inversely correlated to BMIz at 18 months of age and could be viewed as part of a healthy diet. Overweight at 18 months of age was inversely correlated to breastfeeding duration. Finally, this study showed that high maternal BMI was the variable with the strongest correlation to BMIz and overweight in children at 18 months of age.

## Figures and Tables

**Figure 1 nutrients-13-02265-f001:**
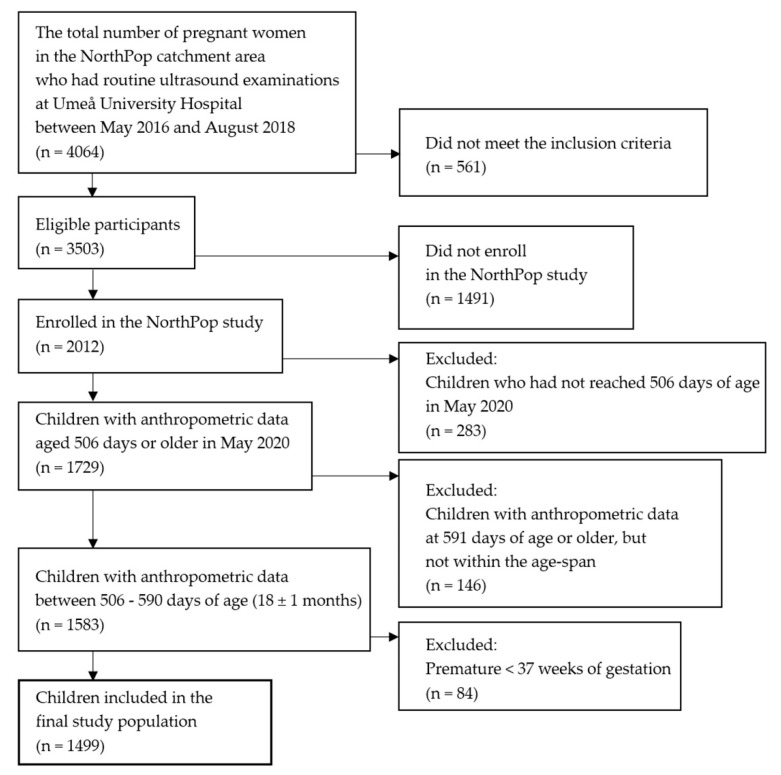
Flow chart illustrating the study population.

**Figure 2 nutrients-13-02265-f002:**
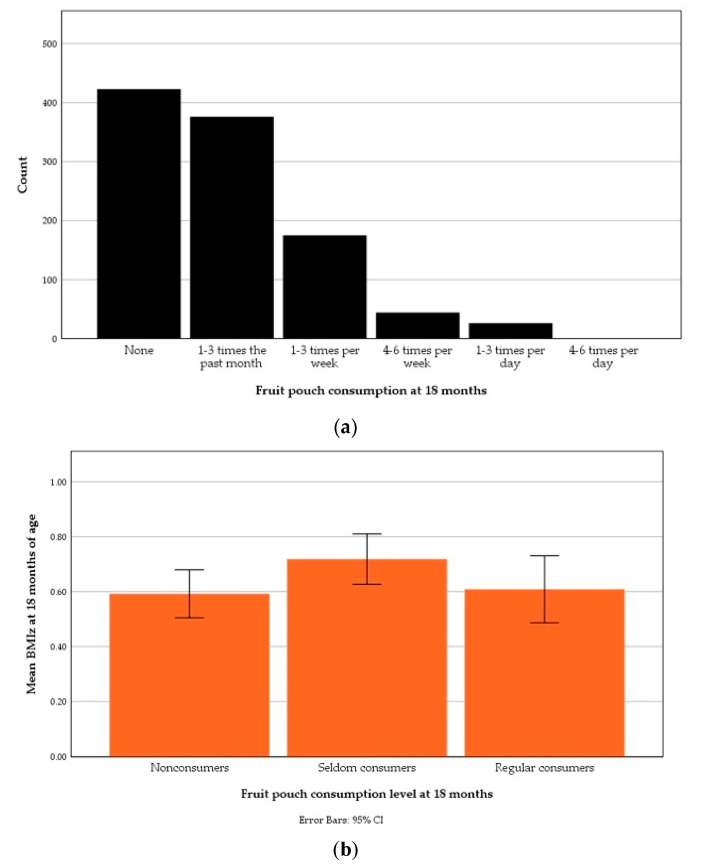
Fruit pouch consumption during the past month at 18 months of age: (**a**) Distribution according to one-month-recall food frequency questionnaire responses; (**b**) An error bar graph illustrating mean body-mass index *z*-score (BMIz) at 18 months of age according to consumption level.

**Figure 3 nutrients-13-02265-f003:**
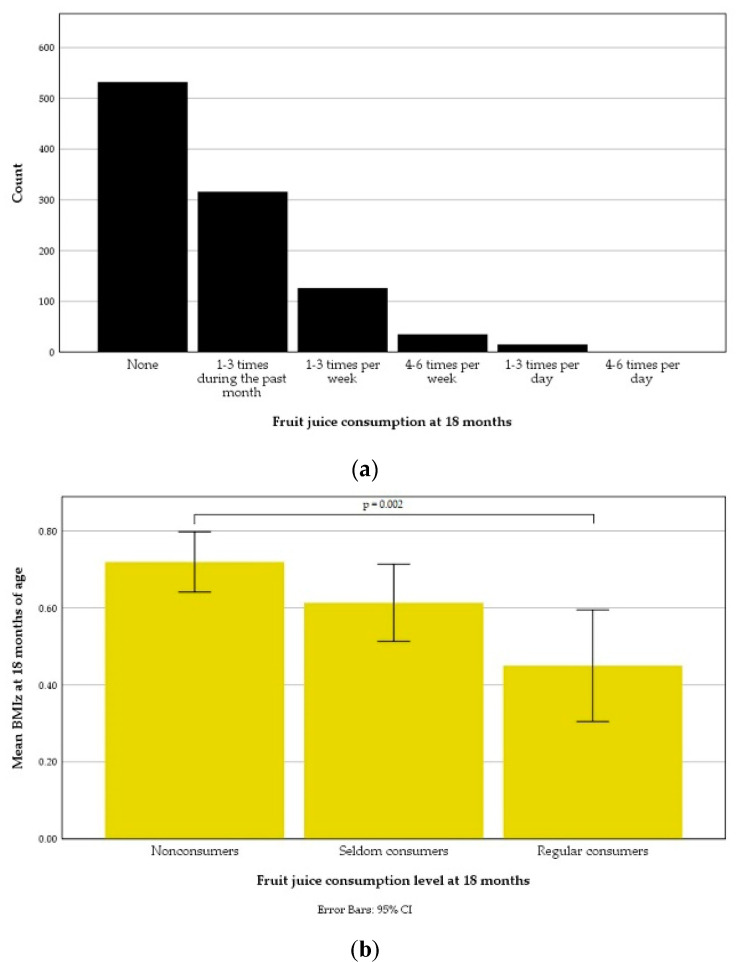
Fruit juice consumption during the past month at 18 months of age: (**a**) Distribution according to one-month-recall food frequency questionnaire responses; (**b**) An error bar graph illustrating mean body-mass index *z*-score (BMIz) at 18 months of age according to consumption level.

**Figure 4 nutrients-13-02265-f004:**
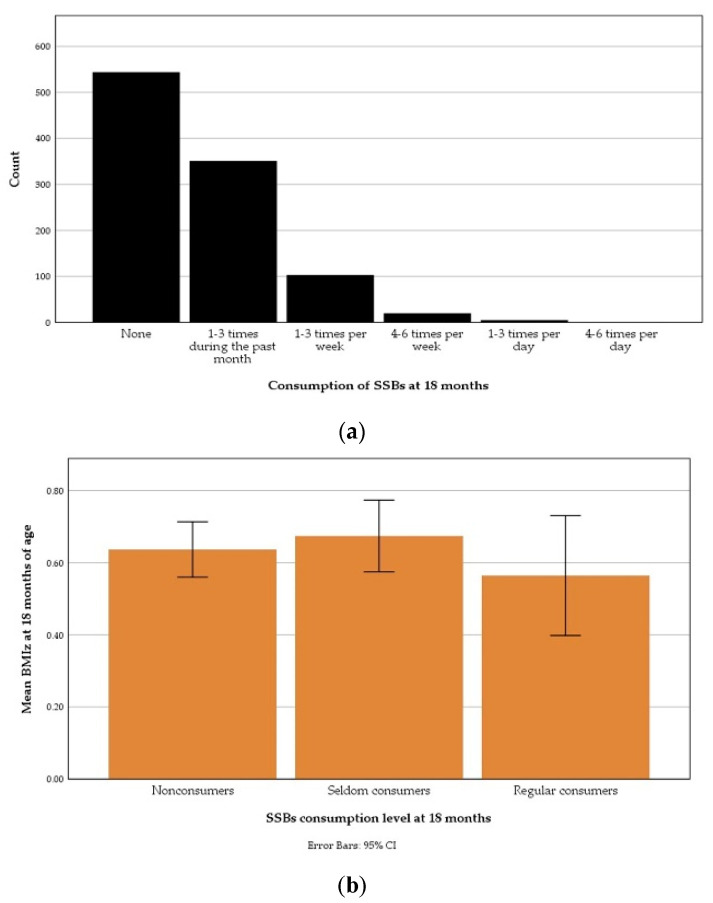
Sugar-sweetened beverage consumption during the past month at 18 months of age: (**a**) According to one-month-recall food frequency questionnaire responses; (**b**) An error bar graph displaying mean body-mass index *z*-score (BMIz) at 18 months according to consumption level.

**Figure 5 nutrients-13-02265-f005:**
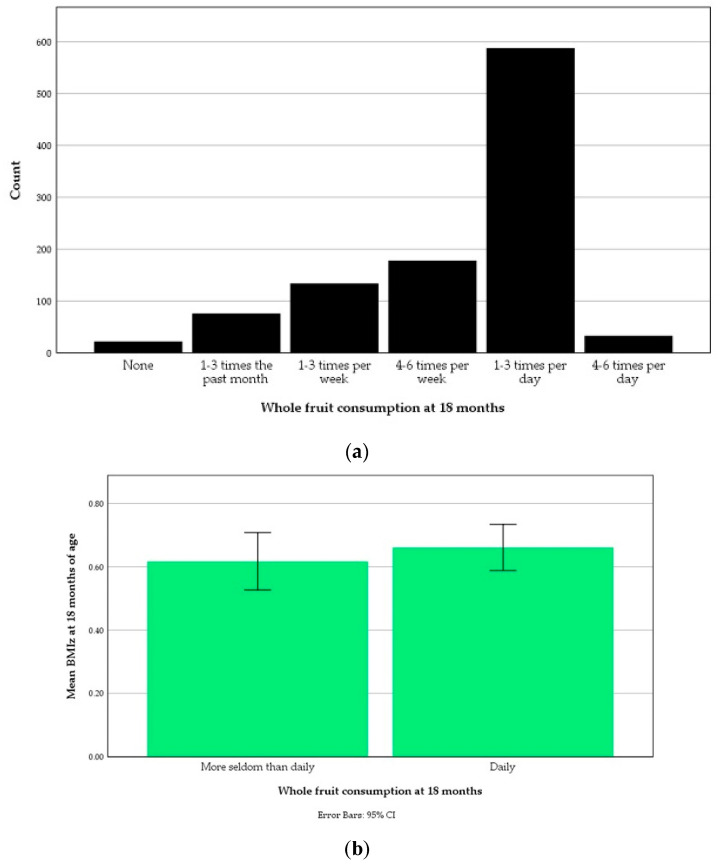
Whole fruit consumption during the past month at 18 months of age: (**a**) Distribution according to one-month-recall food frequency questionnaire responses; (**b**) The means body-mass index *z*-score (BMIz) at 18 months of those who did not consume whole fruit daily and of those who did.

**Figure 6 nutrients-13-02265-f006:**
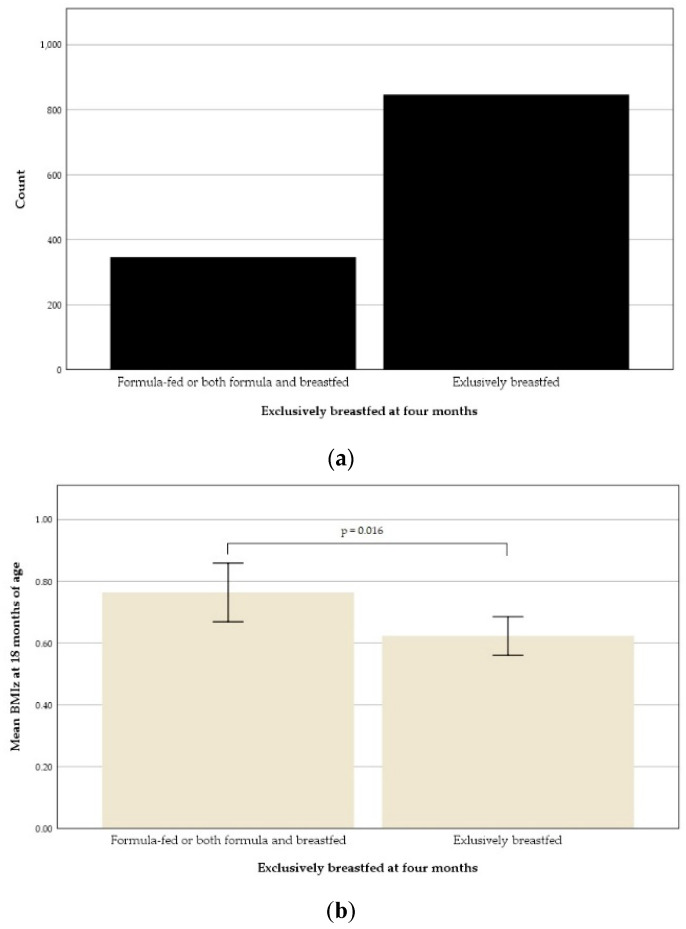
Exclusive breastfeeding at four months of age: (**a**) Distribution shown as block graph; (**b**) An error bar graph illustrating the mean body-mass index *z*-score (BMIz) at 18 months of the formula fed or both formula and breastfed and of the exclusively breastfed children.

**Figure 7 nutrients-13-02265-f007:**
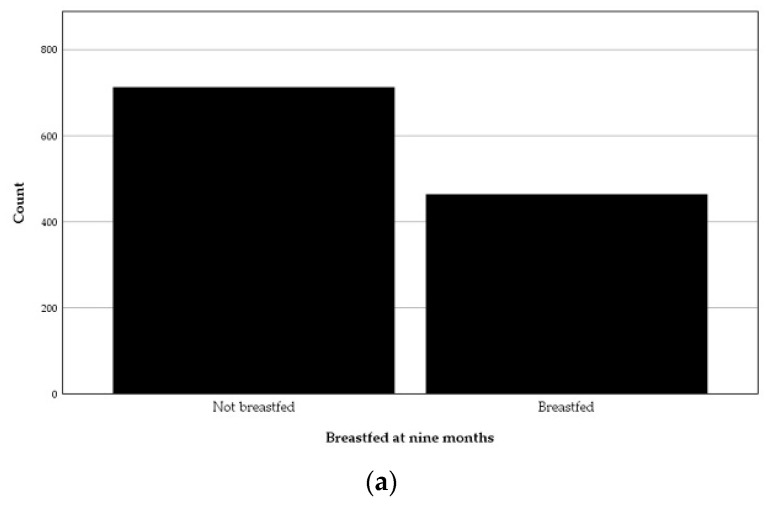
Breastfeeding at nine months of age: (**a**) Distribution; (**b**) An error bar displaying the mean body-mass index *z*-score (BMIz) at 18 months of the *not breastfed* and the *breastfed* children at nine months of age.

**Figure 8 nutrients-13-02265-f008:**
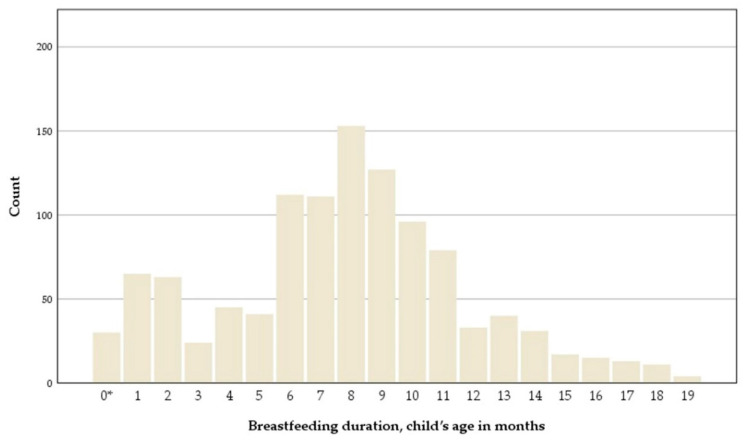
Histogram of breastfeeding duration. * Never breastfed.

**Figure 9 nutrients-13-02265-f009:**
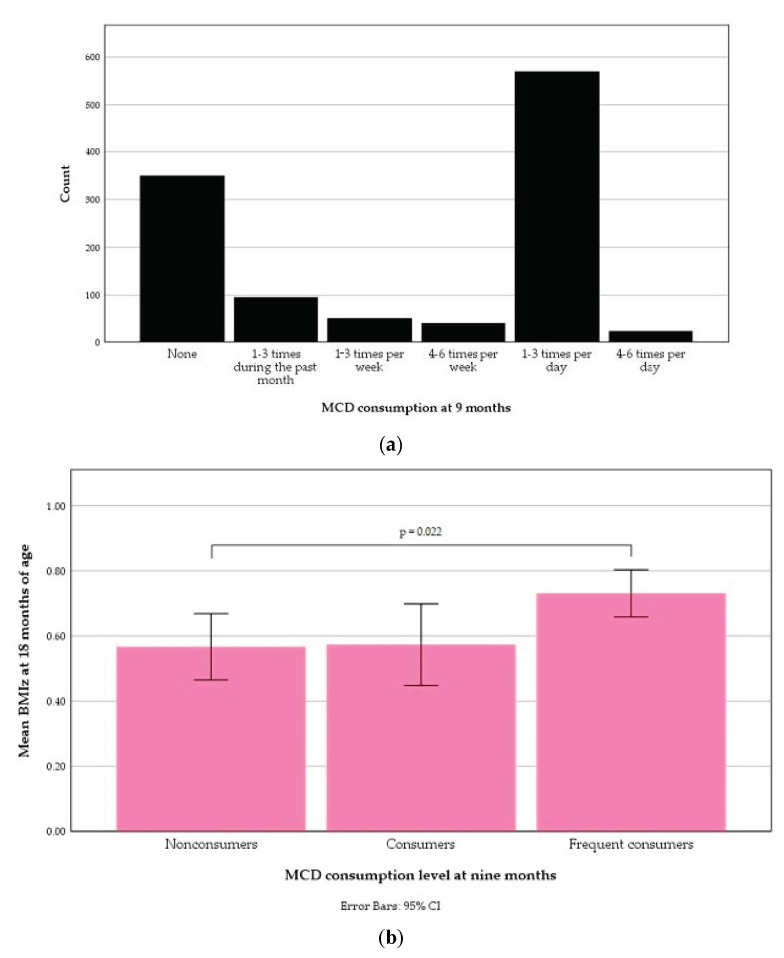
Milk cereal drink (MCD) consumption at nine months of age: (**a**) Distribution according to one-month-recall food frequency questionnaire responses; (**b**) An error bar of distribution in relation to body-mass index *z*-score (BMIz) at 18 months.

**Figure 10 nutrients-13-02265-f010:**
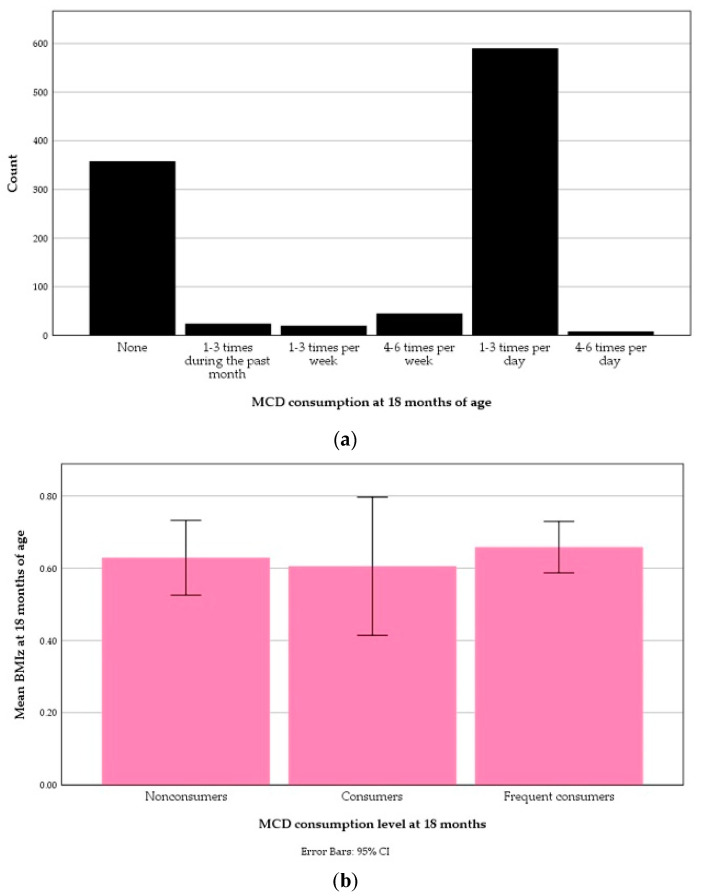
Milk cereal drink (MCD) consumption at 18 months of age: (**a**) Distribution; (**b**) An error bar of distribution in relation to body-mass index *z*-score (BMIz) at 18 months.

**Table 1 nutrients-13-02265-t001:** Characteristics of the participating children and mothers.

	Mean ± SD	*n*	% of Population
**Year of birth**			
2016		155	10.3
2017		723	48.2
2018		621	41.4
**Sex**			
Female		736	49.1
Male		763	50.9
**Gestational length**			
Weeks	40.0 ± 1.3	1499	100
Full term ^1^		1413	94.3
Postterm ^2^		86	5.7
**Weight at birth**			
Grams	3594 ± 478	1499	100
Low birth weight ^2^		10	0.7
Normal weight		1437	95.9
Macrosomia ^3^		52	3.5
**Mothers**			
Age (years)	30.9 ± 4.5	1499	100
Maternal BMI ^4^	24.6 ± 4.2	1333	88.9
GWG ^5^ (kilograms)	15.2 ± 5.9	1197	79.6
**Outcome: BMIz and weight at 18 months of age**			
Age (in days)	552 ± 16	1499	100
BMIz ^6^	0.64 ± 0.93	1499	100
Underweight		3	0.2
Normal weight		1394	93
Overweight ^7^		102	6.8

^1^ Children born preterm were excluded. ^2^ Low birth weight = birthweight < 2500 g. ^3^ Macrosomia = birthweight > 4500 g. ^4^ BMI = body-mass index, in early pregnancy. ^5^ GWG = gestational weight gain. ^6^ BMIz = BMI-for-age *z*-score. ^7^ Including *Obese* (*n* = 6; 0.4%).

**Table 2 nutrients-13-02265-t002:** Parents’ level of education.

Level of Education	Mother		Co-Parent ^1^	
	*n*	%	*n*	%
Primary	10	0.7	6	0.4
Lower secondary	40	2.7	45	3.0
Upper secondary, 2Y ^2^	42	2.8	92	6.1
Upper secondary, 3Y ^3^	321	21.4	479	32.0
Post-secondary < 3Y ^4^	197	13.1	185	12.3
Post-secondary ≥ 3Y ^5^	833	55.6	503	33.6
Post-graduate ^6^	46	3.1	40	2.7
Missing	10	0.7	149	9.9
**Total**	1499	100	1499	100

^1^ Co-parent: father, co-mother or co-parent [42]. ^2^ Two years. ^3^ Three years. ^4^ Shorter than three years. ^5^ Three years or longer. ^6^ Including doctoral degree.

**Table 3 nutrients-13-02265-t003:** Multivariate model for variables associated with body-mass index *z*-score (BMIz) at 18 months of age.

Variables	df	F	Partial η^2^	*p*
Fruit juice consumption at 18 months	2	5.38	0.011	0.005
Breastfeeding at 9 months, yes/no	1	11.78	0.012	0.001
Maternal BMI ^1^	1	20.80	0.022	<0.001

^1^ BMI = body mass index, in early pregnancy.

## Data Availability

The data presented in this study are available from the corresponding author on request. Data are handled in accordance with the General Data Protection Regulation (GDPR) and are therefore not publicly available. The institutional review board and informed consent of the parents permit only the use of aggregated data to be published or made publicly available and prohibit sharing individual data.

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
