# Peer review of "Fruit Pouch Consumption and Dietary Patterns Related to BMIz at 18 Months of Age"

_nutrients, 2021, doi:10.3390/nu13072265_

Round 1
Reviewer 1 Report
Dear Authors,
Thank you for preparing this manuscript. It is a very interesting topic and one that is hardly ever discussed. Many parents rely on fruit pouches for baby weaning. I agree with the facts you included in the manuscript. Maybe you could also add that the effect of the processing of fruit in vitamins and minerals content and also the effect on taste preferences for children.
Author Response
Dear Reviewer,
We thank you for reading our paper and appreciate your input. We are pleased that you find our manuscript interesting and approve of our methods, results and how they are presented. As you point out, there are other concerns than free sugars and overweight when it comes to fruit pouches. We do mention the concern for sweet taste preference in the introduction but feel that vitamin and mineral content is outside the scope of this study, which is focused on the risk of overweight.
Reviewer 2 Report
Thank you for the opportunity to review this study that looks at several dietary factors and maternal factors and their correlation with BMIz among Swedish toddlers at 18 months of age. This paper adds to the literature on dietary factors and maternal factors that may contribute to childhood overweight and obesity. I have a few suggestions that I believe would strengthen the paper.
Major issues:
Materials and Methods
- Lines 76-77, and 93-97 – which FFQ was used? What is its validity and reliability? In particular, is it valid for this age group? Has it been used in other studies? Just stating FFQ is insufficient, as there are many different FFQs and each of them has different reliability and validity in different populations.
- Lines 128-129 – why was a stepwise procedure used?
Discussion
- There is no discussion of the limitations of FFQs. We know that FFQs have many limitations and these really need to be discussed. As above, there is no mention of which FFQ was used, its validity and reliability. These are essential so that readers can determine the likely accuracy of the reported results
Minor issues:
Introduction
- Line 40 – briefly mention the free sugar recommendations? Not all readers may be familiar with them
- Line 48 – would suggest “consumption of sugar-sweetened beverages”
- Lines 61-63 – what about other social and structural determinants of health and how they are linked to childhood overweight and obesity?
Discussion
- It would be nice to have further discussion of how the high incomes and high education levels in this cohort could have influenced the outcomes, especially the surprising outcome that fruit juice consumption was inversely correlated with BMIz.
Overall this is an interesting study and once the concerns around the FFQ are addressed it will be suitable for publication.
Author Response
Dear Reviewer,
Thank you for your insightful comments and constructive criticism, which have helped to improve the manuscript. Please, find our responses in the attachment.
